# UDC-SIT: A Real-World Dataset for Under-Display Cameras

**Kyusu Ahn**[1,3]   **Byeonghyun Ko**[2]   **HyunGyu Lee**[2]   **Chanwoo Park**[2]   **Jaejin Lee**[1,2]

[1]Dept. of Data Science, Seoul National University, Seoul, Republic of Korea
[2]Dept. of Computer Science and Engineering, Seoul National University, Seoul, Republic of Korea
[3]Research Center, Samsung Display Co., Ltd., Yongin, Republic of Korea
kyusu.ahn@snu.ac.kr, {byeonghyun, hyungyu}@aces.snu.ac.kr, {99chanwoo, jaejin}@snu.ac.kr

## Abstract

Under Display Camera (UDC) is a novel imaging system that mounts a digital camera lens beneath a display panel with the panel covering the camera. However, the display panel causes severe degradation to captured images, such as low transmittance, blur, noise, and flare. The restoration of UDC-degraded images is challenging because of the unique luminance and diverse patterns of flares. Existing UDC dataset studies focus on unrealistic or synthetic UDC degradation rather than real-world UDC images. In this paper, we propose a real-world UDC dataset called UDC-SIT. To obtain the non-degraded and UDC-degraded images for the same scene, we propose an image-capturing system and an image alignment technique that exploits discrete Fourier transform (DFT) to align a pair of captured images. UDC-SIT also includes comprehensive annotations missing from other UDC datasets, such as light source, day/night, indoor/outdoor, and flare components (e.g., shimmers, streaks, and glares). We compare UDC-SIT with four existing representative UDC datasets and present the problems with existing UDC datasets. To show UDC-SIT's effectiveness, we compare UDC-SIT and a representative synthetic UDC dataset using five representative learnable image restoration models. The result indicates that the models trained with the synthetic UDC dataset are impractical because the synthetic UDC dataset does not reflect the actual characteristics of UDC-degraded images. UDC-SIT can enable further exploration in the UDC image restoration area and provide better insights into the problem. UDC-SIT is available at: https://github.com/mcrl/UDC-SIT.

## 1 Introduction

Under Display Camera (UDC) is a technology designed to place a camera module under the display to use the UDC area as a display space and take pictures when the camera operates. Since a larger screen-to-body ratio is a common consumer demand that leads to a trend toward bezel-less display products [21, 26, 41], such as smartphones, laptops, tablets, and TVs, the display products equipped with UDC will well meet such a market trend. Moreover, video conferencing uses UDC products that enable natural eye-tracking by arranging a camera in the center of the display. However, UDC has some drawbacks of image deterioration problems, such as low transmittance, blur, noise, and flare.

Since the pixels in the micrometer scale diffract the light traveling through the camera lens [35], degrading image quality, UDC displays have a lower pixel density above the camera lens to reduce transmittance loss and diffraction, as shown in Figure 1. However, a lower display resolution in the UDC area affects natural video viewing. Thus, improving the quality of UDC images is a critical problem. Especially UDC image reconstruction models to enable higher pixel densities around the camera are essential to overcome the problem for more comfortable viewing.

37th Conference on Neural Information Processing Systems (NeurIPS 2023) Track on Datasets and Benchmarks.

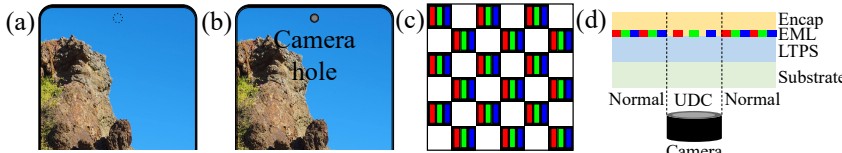

Figure 1: Comparison of conventional hole-display and under-display cameras. The UDC area has a lower pixel density since the pixel pattern functions as diffraction slits. (a) Under Display Camera (UDC). (b) Hole display camera. (c) Pixel structure of the UDC area. (d) Comparison between the UDC area and other display area.

Several studies address the UDC image restoration problem. However, most of them [15, 50, 52] have limitations because their datasets do not completely reflect the properties of actual UDC images (i.e., are synthesized). A challenging problem in constructing the UDC dataset is finding a matching pair of the ground truth and distorted UDC images, which requires significant time and effort.

Feng *et al.*[13] proposes a pseudo-real-world UDC dataset. However, their dataset contains images with occlusions. Ignatov *et al.* [23, 24] try to match two images for the same scene captured by different devices with different positions and angles. They use the SIFT key points [32] of paired images and the RANSAC algorithm [16] to align the two images in a pair. However, the geometric alignment algorithm fails to adequately perform when there is a large discrepancy in the scenes of the two images, such as UDC degradation.

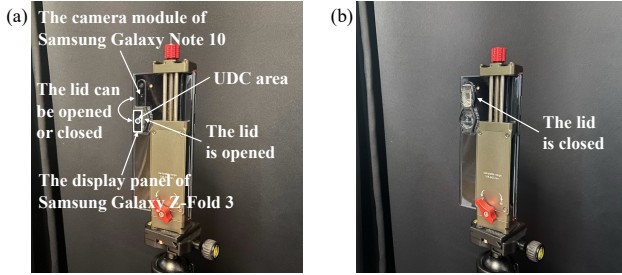

Figure 2: The image capturing system to capture a pair of matching images (the ground-truth image and the degraded image by the UDC). (a) The lid opens to get a ground-truth image. (b) The lid closes to get the corresponding degraded image.

This paper proposes a new UDC dataset called UDC-SIT (UDC's Still Images by the Thunder Research Group). As far as we know, it is the first real-world UDC dataset to overcome the problems of the existing UDC datasets.

We devise an image-capturing system that minimizes lens movement when capturing matching image pairs. We cut and take the UDC area of the flexible display panel of a smartphone with a UDC (e.g., Samsung Galaxy Z-Fold 3 [39]). We use the piece of the display panel as a lid on the camera lens of another smartphone with a non-UDC (e.g., Samsung Galaxy Note 10 [38]). The lid on the non-UDC can be opened and closed. As a result, our image-capturing system has minor geometric inconsistency compared to Feng *et al.* [13]. In addition, it helps to restrict rotations and tilts by the movement of the lid to a degree that does not notably influence alignment. Figure 2 illustrates our image-capturing system. We get a matching pair by capturing the ground-truth image with the lid off and the distorted image with the lid on.

We compensate for the pixel-position difference between the two images for the same scene by exploiting the discrete Fourier transform (DFT) [6]. The discrepancy in pixel positions is inevitable because of the slight movement of the camera when opening and closing the lid.

The contributions of this paper are summarized as follows:

- We propose an image-capturing system for obtaining matched pairs of undistorted and UDC-distorted images for the same scene.
- We propose an image aligning technique for the paired images of the same scene by exploiting the discrete Fourier transform.

- We provide a real-world UDC dataset, UDC-SIT, which accurately reflects actual image degradations by the UDC. UDC-SIT offers a rich set of annotations that stimulate research in UDC image restoration. We make UDC-SIT publicly available (https://github.com/mcrl/UDC-SIT).
- To show UDC-SIT's effectiveness, we compare UDC-SIT and a representative synthetic UDC dataset using five representative learnable image restoration models.

## 2 Related work

**Image restoration.** Image restoration techniques restore a high-quality image from a degraded one. They include denoising, deblurring, and flare removal tasks. Camera-captured images often contain noisy and blurred pixels due to focus errors and incorrect light sensitivity. Restoring these images by removing noise and blur is a common task. Uformer [47] is one of the leading hierarchical transformer-based models for restoring such images. SIDD [1] comprises 30,000 noisy images captured by five smartphone cameras in 10 different scenes under various lighting conditions. DND [34] offers 50 pairs of noisy and noiseless images of very high resolution.

Light flares degrade image quality significantly due to strong light intensities. Wu *et al.* [48] generate synthetic flare removal datasets and train a neural network by modeling the optical characteristics of flares. Also, Dai *et al.* [11, 12] introduce a nighttime flare removal dataset to address the limitations of existing methods that only work well on daytime flares.

The UDC image restoration is complex, and its degradation patterns differ from other restoration tasks of images captured by standard cameras.

**Existing UDC datasets.** Zhou *et al.* [52] tackle the UDC image restoration problem using paired images from a Monitor Camera Imaging System (MCIS) and synthesized PSFs using optical modeling. However, their dataset has limitations, including unrealistic flares captured from a monitor with a limited dynamic range and inaccurate PSFs. They provide only 300 pairs of images for T-OLED and P-OLED, respectively.

Feng *et al.* [15] improve the UDC dataset. To measure the PSF, they place a white point light source one meter away from the OLED display of ZTE Axon 20 [10] and following the methodology presented by Sun *et al.* [43]. By convolving the PSF with HDR images from the HDRI Haven dataset [18], a synthetic dataset is generated. However, it lacks real-world characteristics because the images and PSFs come from different devices. Also, the flare shapes are limited, not including the distorted flares addressed by Yoo *et al.* [50].

Yoo *et al.* [50] introduce a synthetic UDC dataset that includes spatially varying PSFs obtained by optical simulation using the Brown-Conrady Distortion model [46]. However, their simulated flare distortions differ from real-world distortions, and their dataset is not publicly available.

Feng *et al.* [13] create a pseudo-real dataset by capturing degraded images with ZTE Axon 20 UDC [10] and ground-truth images with iPhone 13 Pro camera [25]. They face domain discrepancy and geometric misalignment challenges. Geometric misalignment is severe due to using different camera modules for the two paired images. AlignFormer [13] overcomes the geometric misalignment in UDC by aligning domain information with StyleConv [28] and AdaIN [22], and geometric information with an attention mechanism and a pre-trained optical flow estimator called RAFT [45].

**DNN models for UDC image restoration.** The UDC image restoration challenges [14, 53] use the datasets by Feng *et al.* [15] and Zhou *et al.* [52]. In these challenges, most of the top-ranked teams [31, 44, 49, 54] use the U-Net model [37] and residual networks [19] as their backbone Deep Neural Network (DNN) model for restoration.

## 3 Obtaining aligned images

This section describes our alignment technique of the standard and UDC images for the same scene. Misalignment between the two images is not due to the UDC itself but rather a problem that arises when capturing them. Previous techniques, such as SIFT [32] and RANSAC [16], are inadequate for this purpose because of severe degradation by the UDC. Feng *et al.* [13] use two cameras with

different specifications, leading to variations in perspectives and contents between paired images. Some tools, such as AlignFormer [13], align these images but introduce occlusion regions.

In contrast, our approach minimizes misalignment without introducing occlusion regions. After capturing the two images with our image-capturing system, we exploit discrete Fourier transform (DFT) to align the two paired images. We especially exploit their spatial frequency domain after the DFT to achieve degradation-resilient alignment [9, 17, 27].

## 3.1 Discrete Fourier transform

DFT converts a discrete signal represented by complex exponential waves into constituent frequencies. Equation 1 defines 2D DFT used to obtain the frequency representation of an image.

$$\mathcal{F}(u,v) = \sum_{x=0}^{M-1} \sum_{y=0}^{N-1} f(x,y) \cdot e^{-i2\pi(\frac{ux}{M} + \frac{vy}{N})}, \tag{1}$$

where $\mathcal{F}(u,v)$ denotes the frequency value, $(u,v)$ represents a point in the frequency domain, $M \times N$ is the image size, and $f(x,y)$ is the pixel value at a point $(x,y)$ in the image (i.e., in the spatial domain). Using Euler's formula decomposes the exponential function in Equation 1 into cosine and sine functions, and Equation 1 becomes

$$\mathcal{F}(u,v) = \mathcal{R}(u,v) + i\mathcal{I}(u,v), \tag{2}$$

where $\mathcal{R}(u,v)$ and $\mathcal{I}(u,v)$ denote the real and the imaginary part of $\mathcal{F}(u,v)$, respectively. The *amplitude* $|\mathcal{F}(u,v)|$ and *phase* $\phi(u,v)$ of $\mathcal{F}(u,v)$ are defined as:

$$|\mathcal{F}(u,v)| = \left[\mathcal{R}^2(u,v) + \mathcal{I}^2(u,v)\right]^{\frac{1}{2}} \quad \text{and} \quad \phi(u,v) = \tan^{-1}\left[\frac{\mathcal{I}(u,v)}{\mathcal{R}(u,v)}\right]. \tag{3}$$

## 3.2 Alignment of paired images

Let $M \times N$ be the image size. To measure the difference between the degraded image (D) and ground-truth image (G), we typically use Mean Squared Error (MSE),

$$MSE = \sum_{x=0}^{M-1} \sum_{y=0}^{N-1} (D(x,y) - G(x,y))^2. \tag{4}$$

However, $MSE$ primarily emphasizes local information at points $(x,y)$. In contrast, the spectrum's value for a specific point $(u,v)$ in the frequency domain relies on the collective contribution of all points $(x,y)$ in the spatial domain because $\mathcal{F}(u,v)$ is the sum that iterates through each pixel $(x,y)$ of the image in Equation 1. Thus, to align the two images, we assess the distance between paired images in both spatial and frequency domains as shown in Figure 3. To incorporate both local and global information, we employ a loss function that combines information in the spatial and frequency domains:

$$Loss = \lambda_1 \sum_{x=0}^{M-1} \sum_{y=0}^{N-1} (D(x,y) - G(x,y))^2 + \lambda_2 \sum_{u=0}^{M-1} \sum_{v=0}^{N-1} \Delta\mathcal{F}_{amp}(u,v) + \lambda_3 \sum_{u=0}^{M-1} \sum_{v=0}^{N-1} \Delta\phi(u,v), \tag{5}$$

where $\Delta\mathcal{F}_{amp}(u,v)$ is the L1 distance for the amplitude defined as $\Delta\mathcal{F}_{amp}(u,v) = |\mathcal{F}_D(u,v) - \mathcal{F}_G(u,v)|$, and $\Delta\phi(u,v)$ is the L1 distance for the phase defined as $\Delta\phi(u,v) = |\phi_D(u,v) - \phi_G(u,v)|$).

$\mathcal{F}(u,v)$ at a point $(u,v)$ represents a distinct spatial frequency component, and applying inverse DFT to $\mathcal{F}(u,v)$ at a point $(u,v)$ generates a sinusoidal grating in the spatial domain. Figure 4(a) and Figure 4(c) illustrate the isolation of sinusoidal gratings associated with $\mathcal{F}_G(u,v)$ and $\mathcal{F}_D(u,v)$, respectively.

The significant advantage of using DFT in aligning the original and shifted images lies in its capability to decompose an image into its constituent spatial frequency components. Figure 4(a) and Figure 4(c) display the initial three low-frequency sinusoidal gratings from the two images, clearly revealing a distinct spatial shift. Notably, while the amplitude of sinusoidal gratings remains constant within the same column of Figure 4(a) and Figure 4(c), their corresponding phases exhibit variations. Figure 4(b)

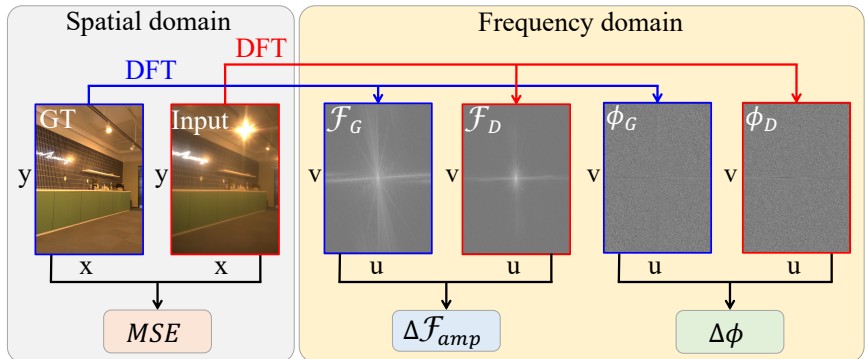

Figure 3: Overview of the proposed loss function to align two paired images. The loss function considers the difference between degraded (D) and ground-truth (G) images in both the spatial and frequency domains.

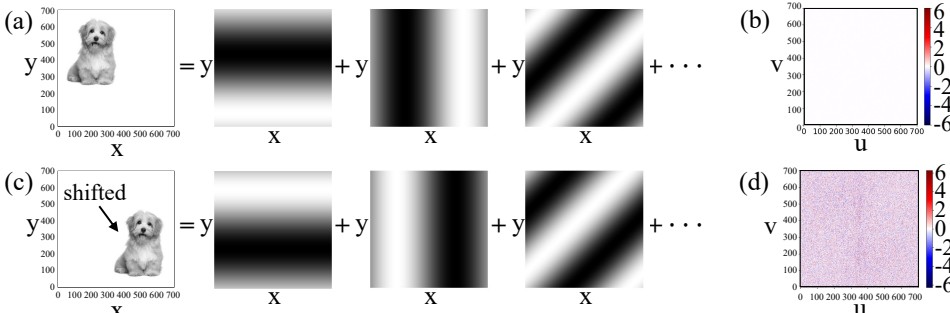

Figure 4: Frequency analysis of two paired images. One is the original image (G), and the other is the spatially shifted image (D) of G. Let $\mathcal{F}_X(u, v)$ be the result of applying $DFT$ to an image $X$. (a) G in the spatial domain comprises multiple sinusoidal gratings. Each sinusoidal grating results from applying inverse $DFT$ to $\mathcal{F}_G(u, v)$. (b) The amplitude difference between G and D. There is no difference. (c) Similar to G, D in the spatial domain comprises multiple sinusoidal gratings. (d) The phase difference between G and D.

and Figure 4(d) visually depict the differences in amplitude (Figure 4(b)) and phase (Figure 4(d)) between the two images. Reducing $\Delta\phi$ is crucial for effectively aligning a misaligned pair of images for the same scene. Figure 4 is a conceptual illustration for paired images involving shifts without degradation. However, due to the degradation, UDC-degraded images also exhibit differences from the ground truth in amplitude. Figure 4 elucidates the importance of the phase component in the alignment of the images.

Our alignment algorithm minimizes the loss in Equation 5 between the ground-truth image and the degraded image. The misaligned degraded image is rotated, shifted and then cropped to achieve the same size as the ground truth until the loss reaches the minimum. Handling rotation and shifts are manageable, but addressing tilt presents a challenge. It is because tilt requires perspective transforms optimized for objects within a single image sharing the same plane. Thus, our current emphasis is on tackling shifts and rotations, with tilt considerations excluded. Nonetheless, our data collection is meticulous, limiting rotations and tilts to a degree that does not notably influence the alignment, as the PCK values in Table 2 affirm.

The original size of camera-captured images is (2016, 1512, 4). The ground truth image is center-cropped to (1792, 1280, 4). The degraded image is similarly cropped around the center. For the degraded image, iterative shifting of (x, y) coordinates and rotation are used to find the minimum loss point where the cropped degraded image aligns with the cropped ground truth image. The final cropped image size becomes (1792, 1280, 4) to ensure H and W are multiples of 256. The detailed algorithm is illustrated in Algorithm 1. Here, we establish $s = 20$, $\theta = 0.3$, $r = 0.1$, and $(\lambda_1, \lambda_2, \lambda_3)$ combinations in Table 3 as hyperparameters.

---

**Algorithm 1** Alignment of paired images $I_G$ and $I_D$

---

**Require:** Images $I_G$, $I_D$ of size $(H, W)$, hyperparameters $s, \theta_r, r, \lambda_1, \lambda_2, \lambda_3$
**Ensure:** Aligned images $C_G$, $C_D$ of size $(H^*, W^*)$
  Crop $C_G$ from $I_G$ using center crop
  Crop $C_D$ from $I_D$ to the size of $C_G$
  Initialize best loss $L_{\text{best}}$ to a large value
  Initialize optimal shifts $s_{\text{opt\_x}}$, $s_{\text{opt\_y}}$, and rotation $\theta_{\text{opt}}$ to 0
  **for** $\theta_{\text{rotation}}$ from $-\theta_r$ to $\theta_r$ with step $r$ **do**
    Apply rotation of $\theta_{\text{rotation}}$ to $I_D$ to get $I_{D_{\text{rotated}}}$
    **for** $x_{\text{shift}}$ from $-s$ to $s$ with step 1 **do**
      **for** $y_{\text{shift}}$ from $-s$ to $s$ with step 1 **do**
        Calculate crop position $(p, q)$ relative to the center crop:
          $p = x_{\text{center\_crop}} + x_{\text{shift}}$
          $q = y_{\text{center\_crop}} + y_{\text{shift}}$
        Crop image $C_{D_{\text{tmp}}}$ from $I_{D_{\text{rotated}}}$ at position $(p, q)$
        Calculate loss $L$ using the loss function in **Eq. 5** between $C_{D_{\text{tmp}}}$ and $C_G$
        **if** $L < L_{\text{best}}$ **then**
          Update $L_{\text{best}}$ to $L$
          Update $s_{\text{opt\_x}}$ to $x_{\text{shift}}$
          Update $s_{\text{opt\_y}}$ to $y_{\text{shift}}$
          Update $\theta_{\text{opt}}$ to $\theta_{\text{rotation}}$
        **end if**
      **end for**
    **end for**
  **end for**
  Apply optimal rotation $\theta_{\text{opt}}$ to $I_D$ to get $I_{D_{\text{rotated}}}$
  Calculate crop position $(p_{\text{opt}}, q_{\text{opt}})$ relative to the center crop:
    $p_{\text{opt}} = x_{\text{center\_crop}} + s_{\text{opt\_x}}$
    $q_{\text{opt}} = y_{\text{center\_crop}} + s_{\text{opt\_y}}$
  Crop $I_{D_{\text{rotated}}}$ to acquire an aligned image $C_D$ at position $(p_{\text{opt}}, q_{\text{opt}})$

---

We capture images in UDC-SIT without any Image Signal Processing (ISP) and in RAW format. While High Dynamic Range (HDR) captures more details in shadows and highlights, authentic UDC images are typically in Low Dynamic Range (LDR). Generating HDR images requires capturing multiple LDR images with different exposures, then combining them to create an HDR image [2]. This process differs from general photography, so we gather our dataset in LDR.

Table 1: Comparison of the UDC datasets. Unlike others, UDC-SIT provides annotations, such as light source, day/night, indoor/outdoor, and flare types. Flare types are classified as shimmer, streak, and glare.

| Dataset | Scene | Dynamic range | Dataset size | Annotations | Publicly available |
|---------|-------|---------------|--------------|-------------|--------------------|
| Zhou *et al.* [52] | Synthetic | LDR | 300 | No | Yes |
| Feng *et al.* [15] | Synthetic | HDR | 2,376 | No | Yes |
| Yoo *et al.* [50] | Synthetic | LDR | - | No | No |
| Feng *et al.* [13] | **Real** | HDR | 6,747 | No | Yes |
| UDC-SIT | **Real** | LDR | **2,340** | **Yes** | Yes |

## 4  Comparison with the existing UDC datasets

In this section, we compare UDC-SIT with the four existing representative UDC image datasets. We summarize the five datasets *Zhou-S* [52], *Feng-S* [15], *Yoo-S* [50], *Feng-R* [13], and *UDC-SIT* in Table 1, where S and R stand for synthetic and real datasets, respectively. Feng *et al.* capture 330 images and then crop them into 6,747 small patches. We explain the detail in the appendix.

**Noises and transmittance decrease.** Under low-light conditions, the camera sensor amplifies both the desired signal and unintended random noise. Since the camera sensor in the UDC is positioned beneath the display pixels that decrease the transmittance, the camera operates in low-light conditions, resulting in noise amplification. The degraded images in the UDC dataset should effectively contain the unique UDC noise, which differs from standard cameras. For example, a

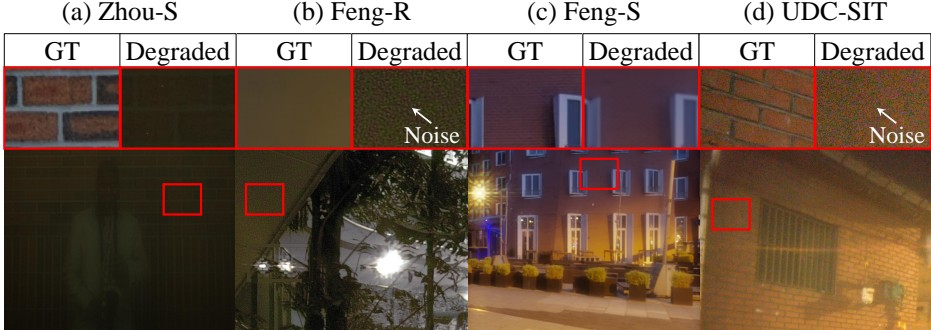

Figure 5: Comparison of the transmittance decrease and UDC noises. GT stands for ground truth. (a) Zhou-S [52]. (b) Feng-R [13]. (c) Feng-S [15]. (d) UDC-SIT.

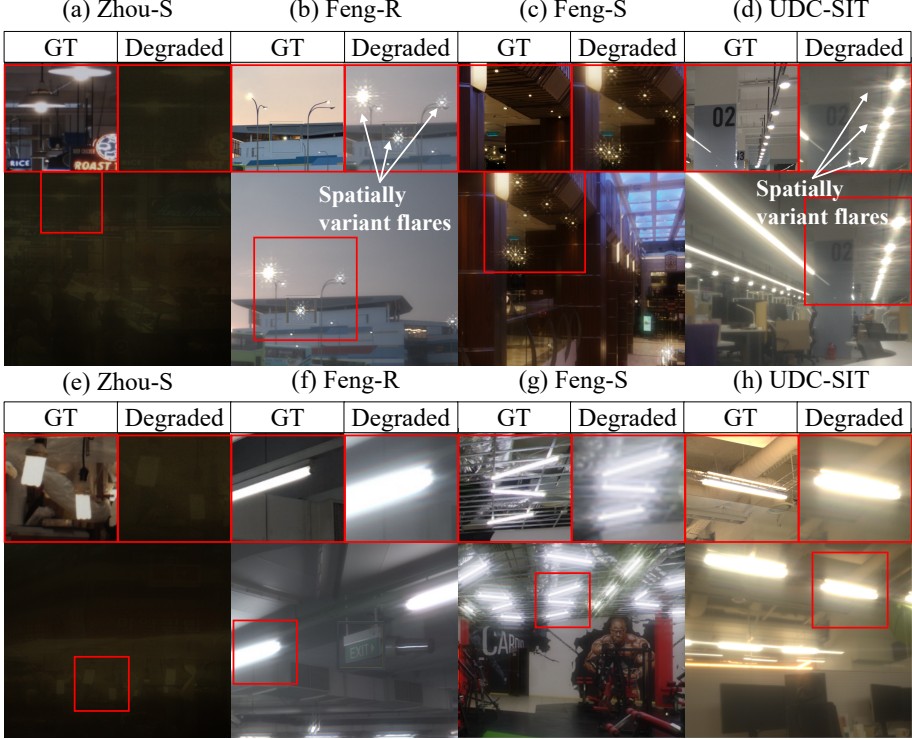

Figure 6: Comparison of flares. The light sources affect the flare shape. The images in (a) Zhou-S [52], (b) Feng-R [13], (c) Feng-S [15], and (d) UDC-SIT show the UDC flares by the round shape of light sources. The images in (e) Zhou-S [52], (f) Feng-R [13], (g) Feng-S [15], and (h) UDC-SIT show the UDC flares by fluorescent light sources.

visual comparison is given in Figure 5. The image in Zhou-S in Figure 5(a) is captured from a monitor under controlled lighting conditions. Thus, there exists no noise. However, we observe the excessive transmittance decrease caused by P-OLED. In addition, the image in Feng-S in Figure 5(c) is generated by convolving the PSF on an HDR image from the HDRI Haven dataset [18]. The HDR images in the dataset are meticulously created through exposure bracketing [3] and cleanup to ensure high quality without overexposure, chromatic aberration, and noise. Thus, the Feng-S image has reduced noise and minimal transmittance loss. On the other hand, the images in Feng-R in Figure 5(b) and in UDC-SIT in Figure 5(d) accurately depict the visible noise caused by the UDC. They also show the effect of transmittance decrease.

Table 2: The PCK comparison between the datasets. UDC-SIT shows the best alignment and has PCK values close to 100% for all values of $\alpha$.

| Dataset | Need alignment | PCK ($\alpha = 0.01$) | PCK ($\alpha = 0.03$) | PCK ($\alpha = 0.10$) |
|---|---|---|---|---|
| Zhou-S [52] | | 98.11 | 98.45 | 99.08 |
| Feng-S [15] | | 99.95 | 99.96 | 99.99 |
| Feng-R [13] | ✔ | **58.75** | **95.08** | **99.93** |
| UDC-SIT | ✔ | **97.26** | **98.56** | **99.35** |

**Flares.** Lens flares typically occur when intense light scatters or reflects within an optical system [11]. In contrast, UDC flares result from light interacting with the display panel above the camera sensor, resulting in undesired reflections, diffraction, or scattering. Thus, it is essential for the images in the UDC dataset to exactly describe real UDC flares if they exist. For example, a visual comparison is given in Figure 6. The images in Zhou-S in Figure 6(a) and (e) do not exhibit any flares since they capture images displayed on a monitor. Similarly, the images in Feng-S in Figure 6(c) and (g) generate unrealistic flares by convolving PSFs to HDR images. Specifically, it lacks the generation of glare, streaks, and spatially variant flares. The shapes of real-world UDC flares vary depending on their positions, resulting in different diffraction patterns. However, the flares in the image in Feng-S in Figure 6(c) appear overly regular. (Figure 6(c) versus (b) and (d)). Moreover, the image in Feng-S in Figure 6(g) demonstrates different fluorescent light flares from the real-world UDC flares in Figure 6(f) and (h). Figure 6(b) versus (d) and (f) versus (h) depict various flare shapes from different UDC smartphones.



Figure 7: Invalid mask visualization in Feng-R [13]. (a) The degraded image captured by the ZTE Axon 20 [10] UDC. (b) The ground-truth image captured by the iPhone 13 Pro rear camera [25]. (c) Aligned ground-truth image by AlignFormer [13].

**Occlusion regions.** Although Feng-R consists of real images, it has some limitations. They avoid capturing close objects to avoid parallax and occlusion. Nonetheless, AlignFormer [13] used in Feng-R creates occlusion regions marked red in Figure 7. This area is typically excluded in training the DNN restoration models. This is why they call Feng-R as pseudo-real pairs.

Table 3: Analyzing UDC-SIT's PCK relative to $\lambda_1$, $\lambda_2$, $\lambda_3$, and $\theta_{\text{rotation}}$. When no rotation is applied, $\theta_{\text{rotation}} = 0$; otherwise, $\theta_{\text{rotation}} = \theta_r$. Notably, $\lambda_2 = 1$ significantly improves the alignment over using MSE alone (i.e., the case of (1, 0, 0, 0)).

| ($\lambda_1$, $\lambda_2$, $\lambda_3$, $\theta_{\text{rotation}}$) | PCK ($\alpha = 0.01$) | PCK ($\alpha = 0.03$) | PCK ($\alpha = 0.10$) |
|---|---|---|---|
| ( 1, 0, 0, 0 ) | 78.77 | 81.09 | 85.65 |
| ( 0, 1, 0, 0 ) | 34.05 | 50.70 | 64.66 |
| ( 0, 0, 1, 0 ) | 62.27 | 64.59 | 72.54 |
| ( 1, 1, 0, 0 ) | 35.55 | 49.41 | 63.25 |
| ( **1**, 0, **1**, 0 ) | **98.22** | **98.73** | **99.31** |
| ( **1**, 1, **1**, 0 ) | **86.13** | **95.02** | **99.23** |
| ( 1, 0.1, 0.1, 0 ) | 72.40 | 77.40 | 83.30 |
| ( 1, 0, 0, 0.3 ) | 43.83 | 47.78 | 59.60 |

**Alignment quality.** Finally, to measure the alignment quality of paired images, we compare the Percentage of Correct Keypoints (PCK) using LoFTR [42] as a keypoint matcher by following the methodology presented by Feng *et al.* [13]. A keypoint pair is considered correctly aligned when $d < \alpha \times max(H, W)$, where $d$ is the position difference between a pair of matched keypoints, $\alpha$ is the

Table 4: Restoration performance for synthetic and real UDC datasets. The term *Input* refers to the PSNR and SSIM values between the paired degraded and ground-truth images.

|  |  | Input | DISCNet [15] | UDC-UNet [31] | Uformer-T [47] | ECFNet [54] | SRGAN [29] |
|---|---|---|---|---|---|---|---|
| Feng-S [15] | PSNR | 26.08 | 43.27 | 49.37 | 42.47 | 52.17 | 32.35 |
|  | SSIM | 0.8561 | 0.9877 | 0.9933 | 0.9844 | 0.9958 | 0.9538 |
| UDC-SIT | PSNR | 21.03 | 26.32 | 27.44 | 27.28 | 28.22 | 24.70 |
|  | SSIM | 0.7330 | 0.8457 | 0.8637 | 0.8594 | 0.9002 | 0.8195 |

threshold, and $H$ and $W$ represent the height and width of the image. To apply a consistent alignment criterion across the datasets with different resolutions, we uniformly set $\max(H, W) = 1024$, aligning with Feng *et al.* [13]. Then the PCK for an image pair is defined by the ratio of the number of correctly aligned keypoint pairs to the total number of keypoint pairs. Table 2 illustrates the comparison result. Since Zhou-S and Feng-S consist of synthetic images, they do not require an additional alignment process, resulting in PCK values close to 100%. On the other hand, Feng-R undergoes alignment using AlignFormer on the images in the paired dataset. It achieves 58.75% for $\alpha = 0.01$. Unlike Feng-R, UDC-SIT consistently exhibits PCK values close to 100% across all values of $\alpha$.

Table 3 shows the analysis result of PCK values on average for various $\lambda_1$, $\lambda_2$, $\lambda_3$, and $\theta_{\text{rotation}}$ combinations. We found that the optimal combination of $\lambda_1$, $\lambda_2$, $\lambda_3$, and $\theta_{\text{rotation}}$ is different for each image pair. The images in UDC-SIT are selected by human inspection, and the PCK values of UDC-SIT in Table 2 originate from the human assessment. The human inspection result is comparable to results obtained by the best combination (1, 0, 1, 0) in Table 3.

Applying rotation (i.e., the case of (1, 0, 0, 0.3)) from Algorithm 1 significantly reduces PCK as shown in Table 3. While this rotation causes a slight MSE decrease, its impact on digitized images leads to notable PCK decline because of changes in ordinary translational correlation during the rotation [7].

To mitigate rotation and tilt effects, we use an image-capturing system in Figure 2, applying only shifts in (x, y) coordinates. Employing the DFT loss without MSE (the cases of (0, 1, 0, 0) and (0, 0, 1, 0)) yields lower PCK values of 34.05% and 62.27%, respectively ($\alpha = 0.01$). Using only MSE and $\Delta\mathcal{F}_{amp}(u, v)$ without $\Delta\phi(u, v)$ (case (1, 1, 0, 0)) results in a reduced PCK of 35.55% ($\alpha = 0.01$). The loss functions encompassing spatial and frequency domains, focusing on $\phi(u, v)$ of $\mathcal{F}(u, v)$ (the cases of(1, 0, 1, 0) and (1, 1, 1, 0)), show high PCK values of 98.22% and 86.13%, respectively ($\alpha = 0.01$). The case of (1, 0.1, 0.1, 0) lacks sufficient $\lambda_2$ and $\lambda_3$ to reflect the frequency domain loss effectively.

## 5 Effects on learnable restoration models

We demonstrate the effectiveness and benefits of UDC-SIT by comparing the UDC image restoration performance with Feng-S. We only compare UDC-SIT to Feng-S because Yoo-S is not publicly available, Feng-R is planned for a future release but is currently unavailable, and Zhou-S needs more flares that are essential for UDC image restoration. We use five learning-based image restoration models, including DISCNet [15], UDC-UNet [31], Uformer [47], ECFNet [54], SRGAN [29] using UDC-SIT. We modify the authors' code of the models to account for the difference in dynamic range and channel size between UDC-SIT and Feng-S [15]. To ensure reproducibility, we explain the experimental setting in the appendix and our GitHub.

The restoration performance of the five models on Feng-S and UDC-SIT is given in Table 4. Models trained with Feng-S successfully restore the images with up to 2.0 times higher PSNR values than Input (except SRGAN). They also achieve a nearly perfect SSIM score. However, none of the models trained with UDC-SIT can restore the images to the same extent as Feng-S. Feng-S inadequately represents actual UDC degradation, such as intense flares. The high degree of glares, shimmers, and streaks results in substantial information loss. Consequently, model performance degradation occurs when those obscured objects become unrecognizable. Thus, restoration becomes more challenging with UDC-SIT than with conventional synthetic datasets.

**Effects of annotations.** It is noteworthy that annotations are specifically available in the UDC-SIT. Flare scenes heavily rely on the capturing environment. For example, artificial lights emit a different

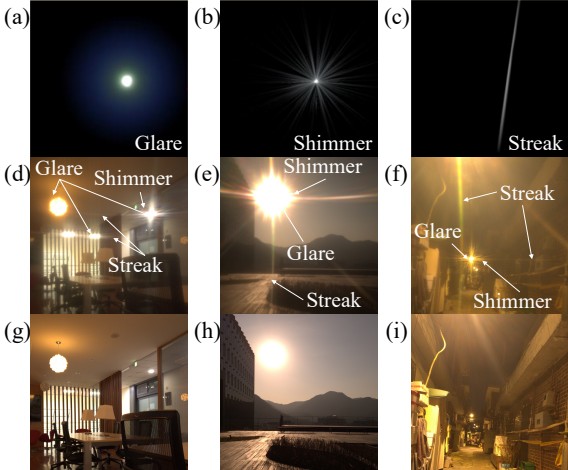

Figure 8: Description of flare components. (a)-(c) The components of a flare include glare, shimmer, and streaks [11]. (d)-(f) Flare components observed in actual degraded images of UDC-SIT. (g)-(i) Ground-truth images corresponding to the UDC degraded images.

spectrum than natural sunlight, resulting in a distinct diffraction pattern. Weak light sources can cause streaks under low-light conditions. Thus, we attach annotations, such as light source, day/night, indoor/outdoor, and flare components, to the images in UDC-SIT. UDC-SIT follows the classification labels of flare components by Dai et al. [11] including *shimmer*, *streak*, *glare*, and *light source* as illustrated in Figure 8. Detailed explanations and instructions for the annotations, and experimental result of their effect on the restoration models can be found in the appendix.

## 6 Limitations

The degradation of images by the UDC depends on the display pixel patterns, which vary between products, as shown in Figure 6. When obtaining UDC-SIT, Galaxy Z Fold 3 [39] display panel is used. Thus UDC-SIT is optimal for restoring Galaxy Z Fold 3 [39] images. Models trained with UDC-SIT may not be suitable for restoring UDC images taken by other devices (e.g., ZTE Axon 20 [10] and Galaxy Z Fold 4 [40]).

## 7 Conclusions

As far as we know, UDC-SIT is the first dataset to include real-world UDC degradation, such as low transmittance, blur, noise, and flare, along with detailed annotations for the light source, day/night, indoor/outdoor, and flare components. With UDC-SIT, one can train a UDC image restoration model to improve the quality of UDC images taken by the UDC. We propose an effective image-capturing system for paired UDC-distorted and ground-truth images. We also propose a technique for aligning the paired images by exploiting the discrete Fourier transform. The experimental result of comparing UDC-SIT and a representative synthetic UDC dataset with five representative learnable restoration models indicates that the models trained with the synthetic UDC dataset are impractical because the synthetic UDC dataset does not reflect the actual characteristics of UDC-degraded images. This implies UDC-SIT is an adequate dataset for UDC image restoration.

## Acknowledgment

This work was supported in part by the Institute for Information & communications Technology Promotion (IITP) grant (No. 2018-0-00581, CUDA Programming Environment for FPGA Clusters) and by the National Research Foundation of Korea(NRF) grant funded by the Korea government(MSIT) (No. RS-2023-00222663). This work was also supported in part by the Samsung Display Co., Ltd. ICT at Seoul National University provided research facilities for this study.

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

# A  Background

In this section, we discuss the factors that contribute to the UDC degradation. We also explore the concept of dynamic range and tone mapping. Also, we discuss color representation and image formats to elucidate the rationale behind collecting the UDC-SIT dataset in the form of RAW images with an `GRBG` 4-channel Bayer pattern.

## A.1  Image degradation due to the UDC

The representative reasons for image degradation due to the UDC can be categorized into three classes: *noise*, *low transmittance*, and *flare* caused by the display panel on top of the camera lens. The reasons for the degradation differ depending on various factors, such as display types (T-OLED and P-OLED), pixel designs, the distance from the display to the camera lens, and the lens structure. Qin *et al.* [35] demonstrates that pixel designs affect the diffraction level. The optimal pixel design could improve the image quality of the UDC. However, the design of the pixels needs to consider many factors, not only the degradation by the UDC but also the size and shape of the red, green, and blue sub-pixels. They affect the color intensity and image sharpness.

Under the UDC setting, because the pixels work as if they were slits, as shown in Figure A.1, the light diffracts, and this causes the degradation. The light diffracts when it passes through obstacles comparable in size to its wavelength. Unfortunately, we cannot avoid diffraction because the aperture size of the pixel layout is on the order of the wavelength of visible light. The diffracted light propagates to the camera sensor through the camera lens.

The Point Spread Function (PSF) of an optical system is the image-irradiance distribution or impulse response on an image plane that results from a point source [5]. For example, an image of a distant star through a telescope is a PSF. It is an energy distribution pattern. It peaks in the center and has characteristics of long-tail low-energy sidelobes. Since a different light source has a different PSF, the flare shape might differ depending on the light source. Also, the diffraction levels of red, green, and blue waves differ because of their wavelengths in the visible ray region.

## A.2  Dynamic ranges and tone mapping

An image's dynamic range describes a scene's luminance range or the limits of the luminance range that a digital camera can capture [33]. The digital camera sensor is narrower than the range of brightness that the human eye can accommodate. Multiple frames of the same scene captured with different exposures, shutter speeds, and F-numbers can be combined to create an image with a higher dynamic range than individually captured frames.

Tone mapping is required to view a video or image captured with High Dynamic Range (HDR) on a monitor. Since the display device has a limited dynamic range unsuitable for reproducing the full range of light intensities in an actual scene, tone mapping approximates the appearance of an HDR image in a medium with a more limited dynamic range by mapping one set of colors to another. One of the representative tone mapping operations is Reinhard tone mapping [36].

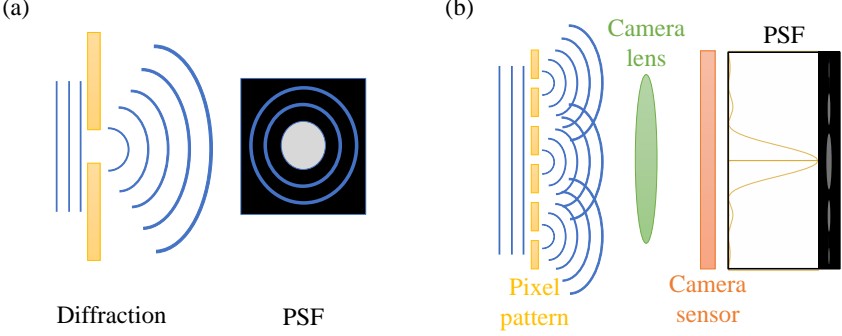

Figure A.1: (a) The diffraction. (b) The PSF is attributed to the pixel pattern on the UDC area.

## A.3 Color representation and image formats

A Bayer pattern [4] is a Color Filter Array (CFA) for arranging RGB color filters on a square grid of photosensors. It uses a checkerboard arrangement of filter patterns, such as `BGGR`, `RGBG`, `GRBG`, or `RGGB`. Image sensors for digital cameras typically use the Bayer pattern. Since each pixel provides only a single-color pixel value (e.g., green) obtained from a CFA sensor, *demosaicing* is used to reconstruct the full-color RGB image. Interpolation computes the pixel's missing color values (e.g., red and blue) using the colors of neighboring pixels.

RAW image files contain unprocessed or minimally processed data from a digital camera's image sensor. RAW image files in recent smartphone camera sensors have a 10-bit range, and each pixel records one of the three colors: red, green, and blue in 1,024 steps using the Bayer pattern. Other image formats, such as JPEG and PNG, are converted from the RAW format using an Image Signal Processor (ISP) with some loss of information in 256 steps. Thus, collecting the images in the RAW format is essential to ensure that the dataset is free from the influence of the ISP (i.e., free from the loss of information).

# B Details of the UDC datasets and DNN models used

This section provides detailed information on the UDC datasets and the DNN models used to compare the datasets in the paper for reproducibility.

The *datasheets for datasets* describe UDC-SIT's hosting, licensing, and maintenance plan. The UDC-SIT dataset, code, and evaluation procedure can be found and downloaded at: https://github.com/mcrl/UDC-SIT.

## B.1 UDC datasets

The images in UDC-SIT are captured in the RAW format using the `GRBG` 4-channel Bayer pattern, effectively eliminating the influence of the built-in ISP. Although HDR captures the details in shadows and highlights better, actual UDC images are predominantly in LDR. Consequently, we capture images in LDR to align with real-world conditions.

We compare the available UDC dataset in Table B.1. Unlike Table 1, it emphasizes information relevant to the DNN model execution. Zhou *et al.* [52] also obtain 16-bit RAW sensor data but only release the paired RGB data. Thus, we classify Zhou-S as 3-channel RGB images. When working with HDR images, tone mapping becomes necessary. Feng *et al.* capture 330 images with dimensions $3200 \times 2400 \times 3$. These are then cropped into 6,521 training images of size $512 \times 512 \times 3$ and 226 test images sized $1024 \times 1024 \times 3$. We provide the dataset in the NPY format (.npy) to make UDC-SIT open and widely used.

## B.2 DNN models for evaluating UDC-SIT

The DNN models used for evaluating UDC-SIT include DISCNet [15], UDC-UNet [31], Uformer-T [47], ECFNet [54], and SRGAN [29]. While we primarily adhere to the original authors' code for the models, we make some modifications.

Table B.1: Comparison of the available UDC datasets.

|  | Zhou-S [52] | Feng-S [15] | Feng-R [13] | UDC-SIT |
|---|---|---|---|---|
| Raw sensor data |  |  | ✔ | ✔ |
| Color representation | RGB | RGB | RGB | GRBG Bayer |
| Dimensions | [1024, 2048, 3] | [800, 800, 3] | [512, 512, 3], [1024, 1024, 3] | [1792, 1280, 4] |
| Dynamic range | LDR | HDR | HDR | LDR |
| Tone mapping |  | ✔ | ✔ |  |
| Value range | [0, 255] | [0, 500] | [0, 255] | [0, 1023] |
| File format | .png | .npy | .png | .npy |

DISCNet, UDC-UNet, and ECFNet are designed to restore UDC images in HDR (i.e., Feng-S [15]), while Uformer-T is designed to restore motion-blurred images in LDR. Also, SRGAN is designed to super-resolve at large upscaling factors while also taking into account the preservation of high-frequency details. Consequently, we make some necessary adjustments to the PyTorch DataLoader, implementing tone mapping [36] for Feng-S and opting for normalization instead of tone mapping for UDC-SIT. Also, we increase the channel size of the models to 4 to match the channel size of UDC-SIT. Training is performed with four NVIDIA GeForce RTX 3090 GPUs. The following are the training details specific to each model for the UDC-SIT dataset:

- **DISCNet.** Due to the channel size of 4 in UDC-SIT, we solely utilize L1 loss as the loss function, excluding perceptual loss. To accelerate training, the models are trained using PyTorch Distributed Data Parallel (DDP) [30] with a batch size of 32.

- **UDC-UNet.** The model output is clamped to the range [0, 1] to calculate the training loss. The additional modification aligns with the details described in DISCNet.

- **Uformer-T.** To maintain consistency with other models operating solely in FP32, we have modified Uformer-T to use FP32 instead of mixed precision. Also, we utilize Transformer feature channels with $C = 8$. The patch size is set to $1,280 \times 1,280$, and the batch size is 4.

- **ECFNet.** Since the training script of ECFNet is not publicly available, we follow the method used in MIMOUNet [9] that heavily affected ECFNet. We increase the base channel size of blocks from 24 to 32 to provide eight channels per image channel. ECFNet introduces the progressive training strategy adopted by Restormer [51]. Specifically, the training phase of the network consists of three stages with different patch sizes (e.g., $256 \times 256$, $512 \times 512$, and $800 \times 800$). We modify the batch size of each stage from 3, 1, and 2 to 16, 4, and 4, respectively. In the third stage, we employ activation recomputation [8] to address the memory constraint of NVIDIA GeForce RTX 3090 GPUs. It effectively reduces the peak memory usage from 46GB to 18GB per device, enabling efficient utilization of available GPU memory resources. We do not use external data or adopt model ensemble strategies that the authors of the models describe.

- **SRGAN.** The model output is clamped to the range [0, 1] to calculate the training loss. Due to the channel size of 4 in UDC-SIT, we utilize L2 loss and adversarial loss as the loss function, excluding perceptual loss.

## C  Details of the annotations

This section describes annotation instructions for crowdsourcing and the effect of annotations in the UDC-SIT.

Table B.2: Annotation examples for UDC-SIT. The corresponding annotated images are shown in Figure B.2. "Indoor" and "Outdoor" scenes are labeled 1 and 3, respectively. The labels for "No distinction," "Day," and "Night" are 1, 2, and 3, respectively. For indoor images, if flares occur due to the influence of natural light sources through windows, we label them "Day" and "Night". For indoor scenes without windows, we label them "No distinction". Glares, shimmers, and streaks are labeled 1 if present and 0 otherwise. If there is no flare, it is labeled 0. The occurrences of flares caused by natural sunlight, artificial light, or both are labeled 1, 2, and 3, respectively.

| File | Indoor/Outdoor (1/3) | No distinction/Day/Night (1/2/3) | Glare (0/1) | Shimmer (0/1) | Streak (0/1) | Light sources (0/1/2/3) |
| --- | --- | --- | --- | --- | --- | --- |
| 317.npy | 3 | 3 | 1 | 1 | 0 | 2 |
| 333.npy | 3 | 3 | 1 | 1 | 1 | 2 |
| 677.npy | 3 | 2 | 1 | 0 | 0 | 1 |
| 689.npy | 3 | 2 | 1 | 1 | 1 | 1 |
| 698.npy | 1 | 2 | 1 | 1 | 0 | 1 |
| 903.npy | 1 | 2 | 1 | 1 | 0 | 2 |
| 905.npy | 1 | 1 | 1 | 1 | 0 | 2 |
| 2384.npy | 3 | 2 | 0 | 0 | 0 | 0 |

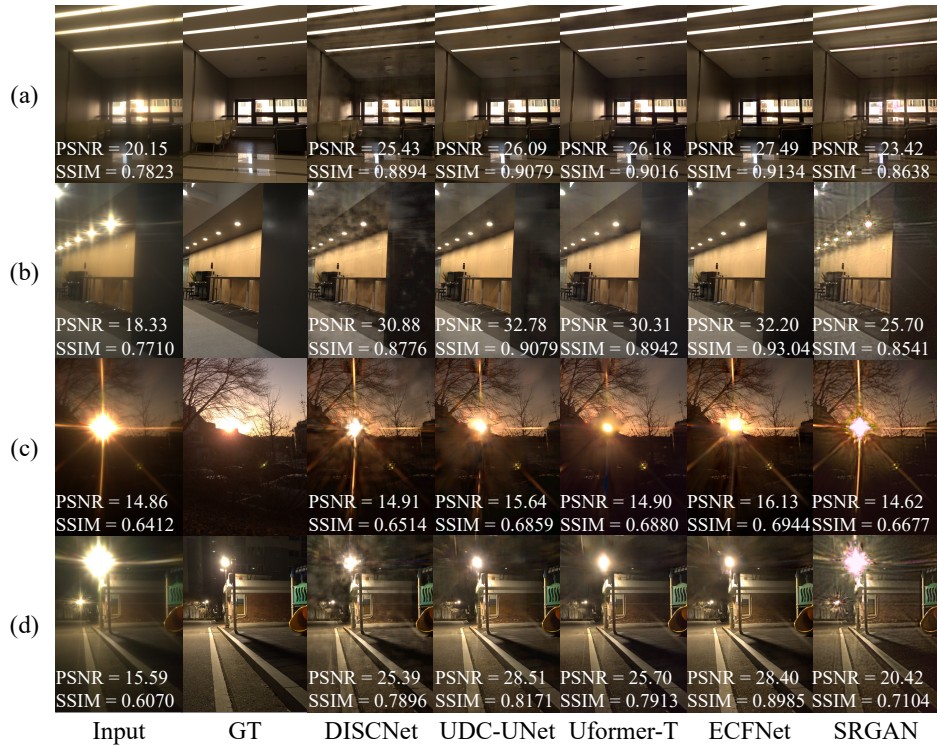

| | | | | | | |
|---|---|---|---|---|---|---|
| Input | GT | DISCNet | UDC-UNet | Uformer-T | ECFNet | SRGAN |

Figure B.1: Visual comparison of flare removal on real-world UDC images. Flare shapes vary according to the captured environments. GT stands for ground truth. (a) Indoor + artificial light (glare + shimmer). (b) Indoor + artificial light (glare + shimmer + streak). (c) Outdoor + day + sunlight (glare + shimmer + streak). (d) Outdoor + night + artificial light (glare + shimmer + streak).

Table B.3: Annotation distribution and the number of pairs.

| Label | # of pairs |
|---|---:|
| Indoor (1) | 1,754 |
| Outdoor (3) | 586 |
| No distinction (1) | 1,340 |
| Day (2) | 649 |
| Night (3) | 351 |
| Glare (0 or 1) | 2,037 |
| Shimmer (0 or 1) | 1,899 |
| Streak (0 or 1) | 1,067 |
| No flare (0) | 273 |
| Natural light (1) | 175 |
| Artificial light (2) | 1,639 |
| Both (3) | 253 |
| **Total** | **2,340** |

## C.1  Effects of image capturing environments

The combination of various factors, including light sources, indoor/outdoor, and day/night, affect the level of degradation or flare shapes in UDC images. The images in Figure B.1(a) show only glare and shimmer, while the images in Figure B.1(b) display glare, shimmer, and streak caused by different artificial light sources. Natural sunlight in Figure B.1(c) has the strongest intensity. It shows extreme flares with glare, shimmer, and streak even in the daytime, while artificial light at night with the low-light condition in Figure B.1(d) has relatively weak intensity but still has extreme glare, shimmer, and streak.

## C.2 Types of annotations

We offer annotations for each image pair. Table B.3 provides a detailed overview of the total count and distribution of different annotation labels. Note that an image pair can have multiple annotation labels. The parenthesized number beside a label is the encoding of the label. The pairs are categorized based on image degradation factors, such as indoor/outdoor, day/night, glare/shimmer/streak, and light sources. If images are captured indoors without any window or access to natural sunlight, there would be no distinction between daytime and nighttime. In such cases, we label them as "No distinction." Detailed information can be found at: https://github.com/mcrl/UDC-SIT.

Participants involved in the crowdsourcing for UDC-SIT are provided with the annotation examples in Table B.2. The corresponding annotated images are shown in Figure B.2. The details of the annotations are described as follows.

**Flare components.**    We follow the classification scheme proposed by Dai *et al.* [11] to categorize different flare components, including glares, shimmers, and streaks. Figure 8 in the main body of the paper and Figure B.2 provide detailed explanations and visual representations of each flare component, serving as comprehensive annotation instructions. A glare refers to the presence of intense and strong light that produces artifacts such as circular light patterns. A shimmer involves rapid and subtle light or color intensity variations within an image. A streak is a long, thin, and

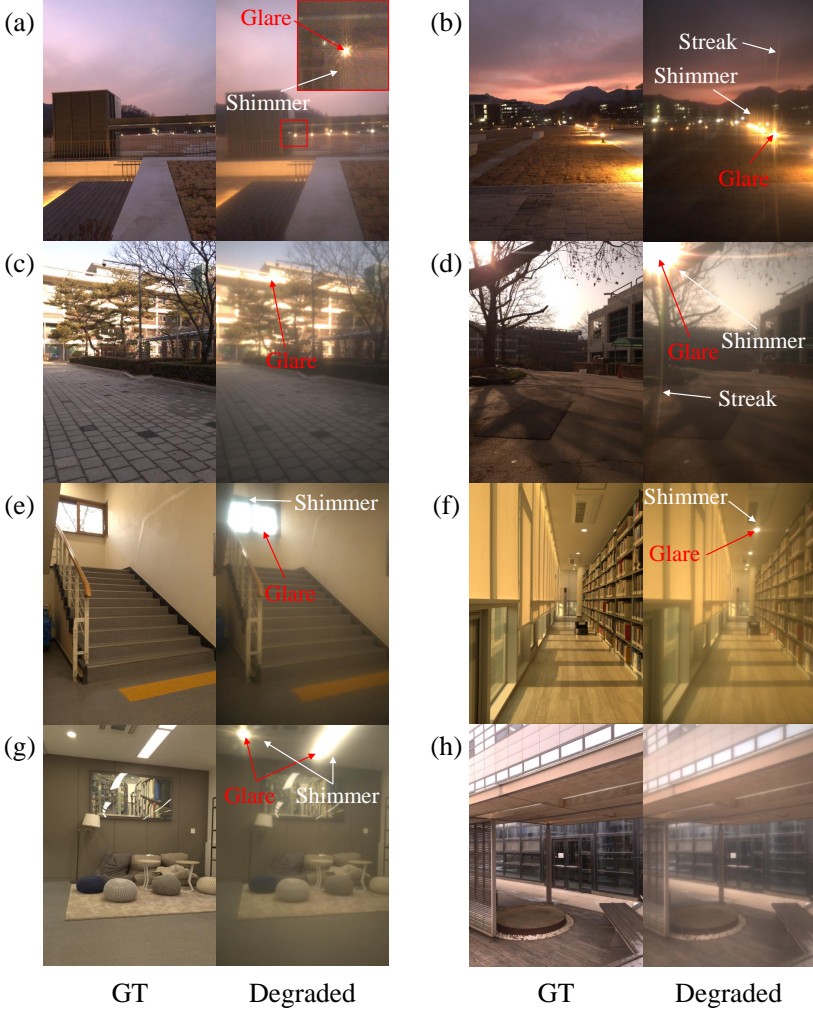

Figure B.2: UDC-SIT in various scenarios. Table B.2 shows the corresponding annotations. (a) 317.npy. (b) 333.npy. (c) 677.npy. (d) 689.npy (e) 698.npy. (f) 903.npy. (g) 905.npy. (h) 2384.npy.

Table C.1: The comparison of restoration performance of the DNN models regarding the annotations in UDC-SIT. "Input" refers to the PSNR and SSIM between the degraded and ground-truth images in UDC-SIT.

|  |  | Input | DISCNet [15] | UDC-UNet [31] | Uformer-T [47] | ECFNet [54] | SRGAN [29] |
|---|---|---|---|---|---|---|---|
| Indoor | PSNR | **21.17** | 26.63 | 27.79 | 27.58 | 28.64 | 25.06 |
|  | SSIM | **0.7437** | 0.8535 | 0.8715 | 0.8680 | 0.9058 | 0.8311 |
| Outdoor | PSNR | **20.53** | 25.18 | 26.13 | 26.15 | 26.66 | 23.37 |
|  | SSIM | **0.6924** | 0.8163 | 0.8346 | 0.8272 | 0.8795 | 0.7761 |
| Day | PSNR | **21.54** | 26.23 | 27.28 | 27.47 | 28.17 | 24.67 |
|  | SSIM | **0.7689** | 0.8669 | 0.8857 | 0.8823 | 0.9152 | 0.8408 |
| Night | PSNR | **19.63** | 25.32 | 26.62 | 25.99 | 27.06 | 23.62 |
|  | SSIM | **0.6012** | 0.7436 | 0.7633 | 0.7536 | 0.8485 | 0.7016 |
| Sunlight | PSNR | **20.48** | 24.49 | 25.52 | 25.62 | 26.22 | 22.72 |
|  | SSIM | **0.7488** | 0.8475 | 0.8670 | 0.8637 | 0.8966 | 0.8147 |
| Artificial light | PSNR | **20.37** | 26.40 | 27.67 | 27.19 | 28.37 | 24.72 |
|  | SSIM | **0.7026** | 0.8279 | 0.8459 | 0.8401 | 0.8895 | 0.8022 |
| No light | PSNR | **22.43** | 26.22 | 26.52 | 27.18 | 27.52 | 24.95 |
|  | SSIM | **0.7180** | 0.8521 | 0.8645 | 0.8623 | 0.8893 | 0.8103 |
| Glare | PSNR | **20.91** | 26.37 | 27.57 | 27.29 | 28.32 | 24.71 |
|  | SSIM | **0.7351** | 0.8456 | 0.8643 | 0.8598 | 0.9024 | 0.8211 |
| Shimmer | PSNR | **20.67** | 26.41 | 27.62 | 27.33 | 28.42 | 24.73 |
|  | SSIM | **0.7301** | 0.8413 | 0.8601 | 0.8557 | 0.9000 | 0.8181 |
| Streak | PSNR | **20.20** | 26.05 | 27.27 | 26.83 | 28.04 | 24.30 |
|  | SSIM | **0.7109** | 0.8266 | 0.8465 | 0.8400 | 0.8897 | 0.8026 |

typically irregular line of light or color observed in an image. Note that the shapes of glares and shimmers vary depending on factors such as day and night conditions, indoor and outdoor settings, and light sources, as illustrated in Figure B.2. In addition, Figure B.2(b) demonstrates that the shape and intensity of glares, shimmers, and streaks differ based on the location within the scene, even when originating from the same light source.

**Light sources.** The intensity and shape of flares vary also depending on the light sources. Flares generated by natural sunlight, as depicted in Figure B.2(d), exhibit an extreme intensity and often include streaks. Additionally, flares can have different characteristics under different conditions, such as day/night and indoor/outdoor settings. For example, the flare in Figure B.2(e) is also caused by natural sunlight, but its shape differs from the flare in Figure B.2(d). Also, Figure B.2(a), Figure B.2(b), Figure B.2(f), and Figure B.2(g) demonstrate variations in the shape of flares caused by different types of lights. Particularly, Figure B.2(b) demonstrates that artificial lights in close proximity during nighttime can induce streaks, and Figure B.2(g) highlights how different types of artificial lights result in various flare characteristics.

**Day/Night.** In the case of indoor images, such as those depicted in Figure B.2(g), the distinction between day and night is generally not required (labeled as "No distinction" in Table B.2). However, we distinguish between day and night when sunlight enters the indoor space through a window, leading to flare occurrence as shown in Figure B.2(e). Indoor images with windows, even if no flare occurs (Figure B.2(f)), are still labeled as day or night. This reflects that windows in indoor spaces may not always cause flares during the day, as flare occurrence can depend on the presence and intensity of sunlight between day and night.

**Indoor/Outdoor.** The illuminance outdoors is generally higher than that indoors during the daytime. Figure B.2(d) and Figure B.2(e) demonstrate that the flares caused by sunlight exhibit variations depending on whether the image is captured indoors or outdoors. Interestingly, the flares in Figure B.2(c) are caused by the reflection of sunlight on the exterior walls of the building.

## C.3 Restoration performance regarding the annotations

Table C.1 shows the comparative analysis of the restoration performance achieved by different DNN models depending on UDC-SIT annotations. Since a UDC-SIT image generally has multiple annotations, the annotation type in a row cannot be regarded as the sole factor influencing UDC degradation when obtaining the PSNR and SSIM values. However, it is reasonable to acknowledge the annotation type as a significant factor that affects the PSNR and SSIM values.

Overall, there is a tendency for the restored image to have low PSNR and SSIM values when the input pair has low PSNR and SSIM values. For instance, Table C.1 demonstrates that input pair captured in outdoor environments exhibit lower PSNR and SSIM values than indoor images. In turn, the restored images by the models also demonstrate inferior PSNR and SSIM values for outdoor restorations compared to indoor restorations. This observation can be attributed to strong sunlight, and intense flares often accompanying outdoor scenes.

The low-light conditions at night influence the degradation significantly. Night conditions tend to be darker, increasing noise and introducing extreme flares. This phenomenon can be observed from the lower PSNR and SSIM values for the annotation Night compared to Day, as indicated in Table C.1.

The presence of flares and the type of light sources inducing flares are also significant factors in UDC degradation. Table C.1 demonstrates the impact of flares on degradation. The input images degraded by sunlight and restored images have low PSNR values compared to the no-light condition. However, interestingly, restored images from the images degraded by artificial light depict higher PSNR values compared to the no-light condition. This tells us that the DNN models are relatively proficient in restoring the degradation caused by artificial light compared to natural sunlight.

Under the sunlight condition, the SSIM values are higher than under the no-light condition. On the contrary, the SSIM values under the artificial light condition are less than those under the no-light condition. PSNR measures the noise level by comparing the original and degraded images. Increasing degradation leads to higher noise and lower PSNR values. On the other hand, SSIM measures structural similarity, considering brightness, contrast, and structural features. Thus, SSIM may exhibit slightly different sensitivity towards different types of degradation compared to PSNR [20]. If degradation factors like flares do not significantly reduce structural similarity, SSIM values will be relatively high.

When flares are present in a degraded image, glares and shimmers appear over a wider area than streaks. Glares and shimmers may significantly impact the reduction of PSNR and SSIM values. On the other hand, streaks typically occur in conjunction with glares and shimmers rather than as a standalone phenomenon. Thus, there may not be significant differences in PSNR and SSIM values based on glares, shimmers, or streaks individually, as shown in Table C.1. However, low-intensity artificial lights at night or natural sunlight in the daytime may cause extreme flares, including streaks. These extreme flares introduce more significant degradation than glares or shimmers alone. Thus, streaks generally exhibit slightly low PSNR and SSIM values compared to glares and shimmers due to their association with extreme flares, as illustrated in Table C.1.

High-resolution images taken by modern smartphones often require training the image-restoration DNN models by patching smaller images. The annotated information, such as indoor/outdoor, day/night, glare/shimmer/streak, and light sources, attached to each patch facilitate the patch-level training in UDC restoration tasks. For example, restoring streaks is challenging for the DNN models when performing the patch-level training, as illustrated in Figure B.1(c). Figure B.1(c) shows big streaks that cover almost the entire image. Providing the annotation information about the entire image can enhance patch-level training by enabling the model to learn correlations between patches and patterns across the patches. This additional information improves the model's ability to restore or enhance images effectively.

In summary, UDC-SIT annotations are crucial for advancing UDC research as none of the existing UDC datasets provide similar information.

