# OpenReview forum: "UDC-SIT: A Real-World Dataset for Under-Display Cameras"
_NeurIPS.cc/2023/Track/Datasets_and_Benchmarks — NeurIPS 2023 Datasets and Benchmarks Poster_

### Official Review · Reviewer_7scG · 2023-07-21
**The website is unavailable**

**Rating:** 6
**Confidence:** 4
**Correctness:** Yes.
**Clarity:** Yes.

**Strengths:**

1. The data collection method is very reasonable.
2. The image alignment technique is novel.
3. The dataset is very challenging for the existing image restoration methods.

**Additional Feedback:**

-

**Documentation:**

No license is available.

**Ethics:**

No.

**Limitations:**

Yes.

**Opportunities For Improvement:**

1. The website is unavailable.
2. After the image alignment, the boundary area of an image would be empty. It is not clear how to handle the empty boundary area.

**Relation To Prior Work:**

Yes.

**Summary And Contributions:**

The authors propose a real-world UDC dataset. In order to collect the dataset, the authors propose an image-capturing system and an image alignment technique that exploits discrete Fourier transform to align a pair of captured images. The experimental results show that this new dataset is challenging.

---

> ### Author Response · Authors · 2023-08-18
>
> Dear reviewer,
>
> Thank you for your useful comments and feedback. We address your questions below:
>
> **Q1: The website is unavailable.**
>  - **(Answer)** Please, check the global official comment.
>
> **Q2: After the image alignment, the boundary area of an image would be empty. It is not clear how to handle the empty boundary area.**
>    - **(Answer)** As illustrated in Section 3.2, the misaligned degraded image is shifted and then cropped to achieve the same size as the ground truth until the loss reaches the minimum. The original size of camera-captured images is $(2016, 1512, 4)$. The ground truth image is center-cropped to $(1792, 1280, 4)$, and the degraded image is similarly cropped around the center. For the degraded image, iterative shifting of $(x, y)$ coordinates is used to find the minimum loss point where the cropped image aligns with the cropped ground truth image. The final cropped image size becomes $(1792, 1280, 4)$ to ensure $H$ and $W$ are multiples of $256$. We present the detailed algorithm in Section 3.2 in the revision.
>
> **Q3: Documentation: No license is available.**
>    - **(Answer)** In the Datasheets for Datasets, we have provided information about our dataset's license and maintenance plan. The license for our dataset is Creative Commons Attribution-NonCommercial-ShareAlike 4.0 International (CC BY-NC-SA 4.0). If the question is about something different, please let us know.

---

> > ### Comment · Reviewer_7scG · 2023-08-25
> >
> > My concerns have been addressed. Thanks.

---

### Official Review · Reviewer_6C5X · 2023-07-21
**UDC paper mentioned flare componts**

**Rating:** 6
**Confidence:** 3
**Correctness:** The evaluation methods are not enough.

**Strengths:**

1. It is interesting to see Under Display Camera (UDC) combine the Flare7k dataset and do the annotation.


**Additional Feedback:**

I am not following the UDC literature, I will look at other reviewer's comments and rebuttals to change my score. I am not confidient in this area.

**Clarity:**

The contributions should rewrite. For example, the first one cannot call a contribution.

**Documentation:**

the provided github link not work

**Ethics:**

no concern

**Limitations:**

1. Four representative learnable image restoration models are not enough. How about GAN-based network, diffusion-based network? I am wondering how they performed compared to transformer-based networks (Uformer).

2. Discrete Fourier transform seems used a lot in image restoration, including UDC. I am wondering is there any differences with the proposed paper and other existing methods.

3. Will the atmosphere also influence the UDC? For example, UDC image under foggy days, will the atmosphere scatter influence lens flare, and make the lens flare more obvious? Have the dataset consider this kind of flare?

**Opportunities For Improvement:**

1. I am wondering is there any relation between UDC and flare. Why PSF synthetic cannot work? It seems models trained on the synthetic dataset Wu et al. [39] work quite well.
2. The images provided are RAW or linear RGB. Will RAW be better than RGB?
3. If people collect the PSF images, convolve with the clean image. Are there any difference with the collected one?
4. Can the author state the difference of model performance on UDC and flare datasset?


**Relation To Prior Work:**

more annotation

**Summary And Contributions:**

1. Under Display Camera (UDC) is an innovative imaging system that places a camera lens beneath a display panel, leading to severe image degradation, including low transmittance, blur, noise, and flare.
2. The paper introduces a real-world UDC dataset named UDC-SIT, which includes non-degraded and UDC-degraded images captured using a proposed image-capturing system and image alignment technique.
3. UDC-SIT contains comprehensive annotations, including light source, day/night, indoor/outdoor, and flare components (shimmers, streaks, and glares), which are absent in other UDC datasets.

---

> ### Author Response · Authors · 2023-08-18
>
> Dear reviewer,
>
> Thank you for your valuable insight and comments. We have addressed the mentioned issues as follows:
>
> **Q1: I am wondering is there any relation between UDC and flare. Why PSF synthetic cannot work? It seems models trained on the synthetic dataset Wu et al. [39] work quite well.**
>    - **(Answer)** As described by Yoo et al. [41], the UDC distortion increases as one moves outward from the center of the lens, resulting in various PSF shapes depending on different image positions. Due to the impossibility of modeling PSF for each location, the generation of synthetic datasets for UDC is inherently limited. The explanation can be found in Section 4 (the flares paragraph) and Figure 6(b, d) of the paper.
>
> **Q2: The images provided are RAW or linear RGB. Will RAW be better than RGB?**
>    - **(Answer)** Every digital camera has an ISP that processes denoising, white balance adjustment, color correction, and brightness correction for RAW images captured under normal conditions. The paper aims to build a dataset suitable for training a DNN model capable of effectively restoring UDC-degraded images that have not undergone ISP processing. Thus, the images are in RAW format. Furthermore, information may also be lost during the demosaicing process of converting 4-channel RAW images to 3-channel RGB images.
>
> **Q3: If people collect the PSF images, convolve with the clean image. Are there any difference with the collected one?**
>    - **(Answer)** Please, see the answer to the previous question about PSF.
>
> **Q4: Can the author state the difference of model performance on UDC and flare datasset?**
>    - **(Answer)** We are not sure whether you are interested in evaluating how UDC DNN models restore a general lens flare dataset (e.g., Flare 7K [7]) or in assessing how regular lens flare restoration DNN models handle the UDC flare dataset (i.e., UDC-SIT). In either case, the unique characteristics of flares caused by UDC, unlike typical lens flares, could result in suboptimal restoration performance.
>
> **Q5: Four representative learnable image restoration models are not enough. How about GAN-based network, diffusion-based network? I am wondering how they performed compared to transformer-based networks (Uformer).**
>    - **(Answer)** We have reviewed and evaluated recent diffusion and GAN-based models suitable for UDC image restoration. For the diffusion model, we evaluate a pre-trained Denoising Diffusion Restoration Model (DDRM) on UDC datasets (Feng-S and UDC-SIT). Unfortunately, the pre-trained model's distribution mismatch hinders successful restoration. While retraining is needed, resource and time constraints prevent us from pursuing it. For GAN-based models, we have included experiments using DCGAN. We are currently conducting an experiment using SRGAN, which has superior performance over DCGAN. This experiment is anticipated to wrap up in the coming days, and subsequently, we intend to substitute DCGAN outcomes with the results from SRGAN next week. We provide the result in Section 5 of the revision.
>
> **Q6: Discrete Fourier transform seems used a lot in image restoration, including UDC. I am wondering is there any differences with the proposed paper and other existing methods.**
>    - **(Answer)** Earlier studies adopt DFT primarily for the loss function during DNN training, particularly for tasks like image restoration. In this paper, we opt for DFT for image alignment. As far as we know, it is the first approach to apply DFT to image alignment.
>
> **Q7: Will the atmosphere also influence the UDC? For example, UDC image under foggy days, will the atmosphere scatter influence lens flare, and make the lens flare more obvious? Have the dataset consider this kind of flare?**
>    - **(Answer)** Weather conditions may affect UDC-captured images. For instance, fog may create glare around light sources through multiple scattering. However, despite distinct physical principles, the glare induced by fog shares common characteristics of lens flares. It suggests the potential feasibility of using the same deep-learning model to restore fog-induced and lens flare. Moreover, UDC is predominantly employed in contexts like video conferencing, where capturing images in weather conditions like fog is less likely. These factors highlight that omitting fog from our dataset does not significantly reduce its value. Nevertheless, weather conditions could still influence images taken with UDC, making it worthwhile to add such scenarios to the dataset as future work.
>
> **Q8: The contributions should rewrite. For example, the first one cannot call a contribution.**
>    - **(Answer)** We appreciate your suggestion. Our intention behind including it was to emphasize the relevance of our paper in the Dataset and Benchmark Track, given the dynamic nature of UDC research. We move it to another part of the introduction in the revision.
>
> **Q9: the provided github link not work**
>  - **(Answer)** Please, check the global official comment.

---

### Official Review · Reviewer_iqzz · 2023-07-21
**Review comments for UDC-SIT**

**Rating:** 6
**Confidence:** 5
**Clarity:** The paper is well-written. The flow o…

**Strengths:**

The paper has a good literature overview of the existing UDC image restoration dataset, provides a comprehensive analysis of existing UDC datasets, and identifies their limitations. The authors devise and propose an image-capturing system to obtain matched pairs of undistorted and UDC-distorted images. Small geometric disparities are further eliminated by the proposed image aligning technique using Discrete Fourier Transform. The proposed real-world UDC dataset, UDC-SIT, accurately reflects actual image degradations by UDC technology, with comprehensive annotations. Comparison of UDC-SIT with a synthetic UDC dataset using four representative learnable image restoration models to demonstrate its effectiveness. The dataset has the potential to inspire further research in the community.

**Additional Feedback:**

See improvement.

**Correctness:**

The construction of the presented dataset is technically sound. The quality of different splits of images is examined by cross-validation to ensure low-quality images are removed.

**Documentation:**

Parts of the dataset hosted on Dropbox are missing. The `.npy` files in the test and validation split are symbolic links and are not available to the reviewer. In addition, the GitHub repo cannot be accessed since it requires verification code to log in.

Apart from that, this work provides the detail of data collection and organization, and availability. It includes documentation and the intended use. The URL for accessing the dataset and Licensing information is provided. The authors are suggested to further describe the hosting and maintenance plan.

**Ethics:**

In this dataset, there are not ethical concerns that warrant further discussion or review.

**Limitations:**

Limitations are discussed in this paper.

**Opportunities For Improvement:**

- In sections 3.1 and 3.2, the authors introduce DFT-based loss. But what is the alignment algorithm? What operations are applied to align the two images?
- As analyzed in Figure 4, there is almost no difference in the amplitude of the two images. Why amplitude distance $\Delta\mathcal{F}_{amp}$ is still added to the loss term? In addition, the analysis and effects of $\lambda_1$, $\lambda_2$, and $\lambda_3$ are missing.
- The authors investigate *shift movement* in frequency domain. But rotation and tilt could be involved in real capturing due to the slight movement of the camera. These movements should also be considered and analyzed.

**Relation To Prior Work:**

This work clearly discusses the difference between the contributions of previous works in the literature.

**Summary And Contributions:**

This paper presented a new large-scale dataset for UDC image restoration, named UDC-SIT. Specifically, the dataset consists of 2,340  images of shape $1792\times1280$. The real UDC dataset overcomes the limitations of existing synthesized datasets. This is made possible by a specifically-devised image-capturing system to obtain matching pairs of undistorted and UDC-distorted images for the same scene. The image alignment technique based on  Discrete Fourier Transform (DFT) helps compensate for minor misalignment between the two images. Experimental analysis shows the property of the proposed dataset, with a comparison to three existing datasets. The main contributions of this work are the proposed dataset and the corresponding evaluations.

---

> ### Author Response · Authors · 2023-08-18
>
> Dear reviewer,
>
> Thank you for your detailed review and useful comments which help us improve our work. We address the key issues as follows:
>
> **Q1: In sections 3.1 and 3.2, the authors introduce DFT-based loss. But what is the alignment algorithm? What operations are applied to align the two images?**
>    - **(Answer)** As illustrated in Section 3.2, the misaligned degraded image is shifted and then cropped to achieve the same size as the ground truth until the loss reaches the minimum. The original size of camera-captured images is $(2016, 1512, 4)$. The ground truth image is center-cropped to $(1792, 1280, 4)$, and the degraded image is similarly cropped around the center. For the degraded image, iterative shifting of $(x, y)$ coordinates is used to find the minimum loss point where the cropped image aligns with the cropped ground truth image. The final cropped image size becomes $(1792, 1280, 4)$ to ensure $H$ and $W$ are multiples of $256$. We present the detailed algorithm in Section 3.2 in the revision.
>
> **Q2: As analyzed in Figure 4, there is almost no difference in the amplitude of the two images. Why amplitude distance $\Delta \mathcal{F}_{amp}$ is still added to the loss term? In addition, the analysis and effects of $\lambda_1$, $\lambda_2$, and $\lambda_3$ are missing.**
>    - **(Answer 1)** Figure 4 is a conceptual illustration for paired images involving shifts without degradation. However, due to the degradation, UDC-degraded images exhibit differences from the ground truth in amplitude. Figure 4 elucidates the importance of the phase component in the alignment of the images. In Table 3 of the revision, $\Delta \mathcal{F}\_{amp}(u,v)$ might appear to hinder alignment, but human scrutiny identifies situations where improved alignment coexists with $\Delta \mathcal{F}\_{amp}(u,v)$ rather than $\Delta\phi(u,v)$. We add this paragraph in Section 3.2 in the revision.
>    - **(Answer 2)** Table 3 in the revision is the analysis for the PCK of UDC-SIT with respect to $\lambda_1$, $\lambda_2$, and $\lambda_3$ and $\theta_𝑟𝑜𝑡𝑎𝑡𝑖𝑜𝑛$. The application of DFT loss without MSE yields notably lower PCK values. Loss functions encompassing both spatial and frequency domains, particularly focusing on the phase $\phi(u,v)$ of $\mathcal{F}(𝑢,𝑣)$, demonstrate the highest PCK. We add the analysis result in Section 4 in the revision.
>
> **Q3: The authors investigate *shift movement* in frequency domain. But rotation and tilt could be involved in real capturing due to the slight movement of the camera. These movements should also be considered and analyzed.**
>    - **(Answer)** As you mentioned, rotation and tilt could be involved in real capturing. Nonetheless, our data collection was meticulous, limiting rotations and tilts to a degree that does not notably influence alignment, as the PCK values in Table 2 affirm. Addressing tilt presents a challenge, requiring perspective transforms optimized for objects within a single image sharing the same plane. Still, evaluating the impact of rotation remains valuable. Section 4 of the revision evaluates and analyzes the alignment result using clockwise and counterclockwise rotations. Considering the complexities related to rotation and tilt, we engineered the capturing system to minimize angle variations in the image pair.
>
> **Q4: Parts of the dataset hosted on Dropbox are missing. The `.npy` files in the test and validation split are symbolic links and are not available to the reviewer. In addition, the GitHub repo cannot be accessed since it requires verification code to log in.**
>    - **(Answer)** Please, check the global official comment.

---

### Official Review · Reviewer_YpcN · 2023-07-26
**A new real-world under-display-camera degradation dataset**

**Rating:** 6
**Confidence:** 4
**Correctness:** Yes.
**Clarity:** Good.

**Strengths:**

1. A new real-world UDC degradation dataset with aligned images.
2. The acquistion equipment is easy yet effective.
3. The alignment technique seems useful for aligning the images with/without UDC degradation.
4. The comparison with prior datasets are detailed and the paper is easy to follow.

**Additional Feedback:**

See the limitation part.

**Documentation:**

Yes, the datasheet for datasets is provided.

**Ethics:**

No.

**Limitations:**

1. It seems that the additional annotations including the type of flares, the light sources are not utilized in this paper. Can authors provide some simple applications with that part of annotations to show its usage and importance compared to other datasets?

2. The proposed datasets is only applicable to specific display type and camera using to collect such a dataset. It is recommended that some generalization experiments are conducted to simply show such issues and give some insights on what needs to be handled in the future.

3. UDC is usually used as the front-camera including smart phones and monitors, as motivated in this paper, is there any plans to include the  selfie scenarios or test on such cases while keeping the privacy unvoilated?

**Opportunities For Improvement:**

See limitations below.

**Relation To Prior Work:**

Yes.

**Summary And Contributions:**

This paper proposes a new real-world UDC degradation dataset by designing a simple yet effective data acquisition equipment to put/remove a screen in front of a camera. A DFT-based image alignment technique is also proposed to align the paired images with and without the occlusion of the camera. Detailed comparison with existing UDC datasets including pesudo-real and synthetic ones are provided and experiments with several image restoration models on the proposed dataset and a competing dataset are performed.

---

> ### Author Response · Authors · 2023-08-18
>
> Dear reviewer,
>
> Thank you for your review and useful insight. We address the individual comments below:
>
> **Q1: It seems that the additional annotations including the type of flares, the light sources are not utilized in this paper. Can authors provide some simple applications with that part of annotations to show its usage and importance compared to other datasets?**
>    - **(Answer)** Simple applications of the annotations can be found in Section 3.3 of the supplementary material. Streaks are mainly caused by artificial light captured at night or natural light captured during the day in outdoor settings. However, restoring these streaks presents a challenge for DNN models during patch-level training of the images. Supplying annotation information for the entire image can enhance patch-level training by allowing the model to learn correlations between patches and patterns across them. Additionally, images without streaks can be relatively easily restored. Consequently, restoring streak-free images using lightweight models in practical mobile applications can reduce inference time.
>
> **Q2: The proposed datasets is only applicable to specific display type and camera using to collect such a dataset. It is recommended that some generalization experiments are conducted to simply show such issues and give some insights on what needs to be handled in the future.**
>    - **(Answer)** As addressed in Section 6 (Limitations), the diversity in flare shapes across different devices caused by various pixel designs renders UDC-SIT less applicable to restoring UDC images from different devices from different vendors. This discrepancy is evident in Figure 6 (c, g) and (d, h) in the paper, showcasing significant variations in flare shapes depending on vendors. When the pixel designs are significantly different from the device used by UDC-SIT, the model trained by UDC-SIT can be used as a pretrained model and fine-tuned with a small dataset for the new device to restore UDC images. However, when the same manufacturer releases a new device (e.g., Samsung Galaxy Z Fold 3, 4, and 5), the flare shape will likely remain the same compared to its predecessors.
>
> **Q3: UDC is usually used as the front-camera including smart phones and monitors, as motivated in this paper, is there any plans to include the selfie scenarios or test on such cases while keeping the privacy unvoilated?**
>    - **(Answer)** Indeed, selfie scenarios occur frequently for smartphones. While avoiding privacy infringements related to human faces, gathering a sufficient dataset containing images of human faces is challenging. However, two aspects can be considered as a potential avenue for future work. First, it is worth exploring whether models trained on UDC-SIT (datasets without human faces) can effectively restore UDC-degraded images containing human faces. Second, supplementing the dataset by obtaining consent and capturing dozens of images from individuals can be considered an approach to enhance the dataset. Again, the UDC-SIT pretrained model can be fine-tuned with the small selfie image dataset.

---

### Author Response · Authors · 2023-08-18
**The revised portions are highlighted in yellow.**

Dear reviewers,

We appreciate your valuable feedback and insightful comments. We have uploaded a revised version of our paper along with supplementary material. The revised portions are highlighted in yellow.

In the case of the dataset link, a symbolic link was initially shared mistakenly. We have now re-uploaded the dataset to the updated link below. Currently, the dataset on GitHub is accessible privately, as it cannot be made publicly available before the NeurIPS conference. Upon acceptance of this paper, we will promptly make it publicly available.
- Dataset link: https://www.dropbox.com/scl/fi/4jtsxjm4xx8q375dt9i9x/UDC-SIT-v2.tar.gz?rlkey=w202pw16w402izohsq2kldpd3&dl=0 (or `wget https://jinpyo.kim/data/UDC-SIT-v2.tar.gz`)
- Dataset file name: `UDC-SIT-v2.tar.gz`
- GitHub ID: `udcsit2023@gmail.com`
- GitHub password: `reviewers@23`
- Once you've received a device verification message upon logging into the mentioned GitHub account, you can use the Gmail account above (same ID and password) to log in Gmail to get `device verification code`. (`ID: udcsit2023@gmail.com`, `password: reviewers@23`)


Thank you for your consideration.

---

### Comment · Area_Chair_PHNu · 2023-08-22
**Author/reviewer discussions**

Dear Reviewers,

Thank you for being a reviewer for NeurIPS 2023 Datasets and Benchmarks!

The authors have already submitted their feedback and I noticed that you don't appear to have submitted a new round of comments.

Could you examine rebuttals and other reviewers' comments, and open up discussions with the authors and other reviewers? Thank you!

Best regards,
Your AC

---

### Author Response · Authors · 2023-08-25
**A new revision has been submitted.**

Dear reviewers,

A new revision has been submitted. In comparison to the previous revision, the new revision appends "_v2" to the filenames of both the paper and supplementary materials. In terms of content, the change made includes the incorporation of results from an experiment where DCGAN was replaced with SRGAN. The modified sections are as follows, and similar to the previous revision, they have been highlighted in yellow:

- Paper Section 5 (including Table 4)

- Supplementary Materials Section 2.2 (including Figure 2 and Table 4)

---

### Decision · Program_Chairs · 2023-09-22

**Decision:**

Accept (Poster)

**Comment:**

The paper introduces a real UDC dataset, which is achieved by an image-capturing system and an image alignment technique. This dataset holds significant importance in the research area. Additionally, the system and alignment methods introduced in the paper offer valuable insights for related research efforts. The experiments conducted in the study are thorough and sufficient.

The consensus among the four reviewers (6,6,6,6) resulted in an average rating of 6 across the board. The primary concerns raised by the reviewers primarily revolved around details, most of which have been adequately addressed by the authors in the updated version and subsequent discussions. While the paper received an average score, the general consensus was that it was marginally above acceptance threshold. Furthermore, the concerns and minor issues could be resolved in the final version of the paper.

Therefore, the AC recommends the paper for acceptance. The feedback provided by the reviewers is valuable and can be used to improve the paper further. It is essential to incorporate all modifications and updates into the final version and ensure consistency in the referencing style, taking into account that some papers have been accepted by journals and conferences, not just in their arXiv versions.